# Sodium Propionate Relieves LPS-Induced Inflammation by Suppressing the NF-ĸB and MAPK Signaling Pathways in Rumen Epithelial Cells of Holstein Cows

**DOI:** 10.3390/toxins15070438

**Published:** 2023-07-03

**Authors:** Chenxu Zhao, Fanxuan Yi, Bo Wei, Panpan Tan, Yan Huang, Fangyuan Zeng, Yazhou Wang, Chuang Xu, Jianguo Wang

**Affiliations:** 1College of Veterinary Medicine, Northwest A&F University, Yangling 712100, China; cxzhao@nwafu.edu.cn (C.Z.); fanxuanyi@nwafu.edu.cn (F.Y.); weibo1110227@nwafu.edu.cn (B.W.); tanpp@nwafu.edu.cn (P.T.); hy0016@nwafu.edu.cn (Y.H.); zengfy@nwafu.edu.cn (F.Z.); wangyazhou@nwafu.edu.cn (Y.W.); 2College of Animal Science and Veterinary Medicine, Heilongjiang Bayi Agricultural University, Daqing 163000, China; 3College of Veterinary Medicine, China Agricultural University, Beijing 100193, China

**Keywords:** sodium propionate, lipopolysaccharide, subacute ruminal acidosis, rumen epithelial cells, inflammation

## Abstract

Subacute ruminal acidosis (SARA) is a prevalent disease in intensive dairy farming, and the rumen environment of diseased cows acidifies, leading to the rupture of gram-negative bacteria to release lipopolysaccharide (LPS). LPS can cause rumentitis and other complications, such as liver abscess, mastitis and laminitis. Propionate, commonly used in the dairy industry as a feed additive, has anti-inflammatory effects, but its mechanism is unclear. This study aims to investigate whether sodium propionate (SP) reduces LPS-induced inflammation in rumen epithelial cells (RECs) and the underlying mechanism. RECs were stimulated with different time (0, 1, 3, 6, 9, 18 h) and different concentrations of LPS (0, 1, 5, 10 μg/mL) to establish an inflammation model. Then, RECs were treated with SP (15, 25, 35 mM) or 10 μM PDTC in advance and stimulated by LPS for the assessment. The results showed that LPS (6h and 10 μg/mL) could stimulate the phosphorylation of NF-κB p65, IκB, JNK, ERK and p38 MAPK through TLR4, and increase the release of TNF-α, IL-1β and IL-6. SP (35 mM) can reduce the expression of cytokines by effectively inhibiting the NF-κB and MAPK inflammatory pathways. This study confirmed that SP inhibited LPS-induced inflammatory responses through NF-κB and MAPK in RECs, providing potential therapeutic targets and drugs for the prevention and treatment of SARA.

## 1. Introduction

In pursuit of higher production performance, high-yielding dairy cows are often fed more concentrated diets during lactation to meet energy requirements [1]. The starch-rich grains are rapidly fermented within the rumen, which is followed by a rapid decrease in rumen pH, eventually leading to subacute ruminal acidosis (SARA) [2]. Cows suffering from SARA will have lower milk production and fat rate, affecting milk quality [3]. Improving milk production as much as possible without causing SARA is the goal for all farmers. The clinical symptoms of SARA are generally lagging and atypical, which makes it difficult to diagnose [4]. Detection of ruminal fluid pH is commonly used to diagnose ruminal acidosis, but sampling and evaluation of rumen fluid are not part of routine testing because it can cause stress to cows and be time-consuming [4,5]. Studies have shown that a long period at a low pH in the rumen will cause gram-negative bacteria to multiply greatly; then the bacteria will lyse, and the concentration of free lipopolysaccharide (LPS) in the rumen increases dramatically [2]. In the case of SARA, the high concentration of LPS and high acidity in the rumen fluid can impair the barrier function of the rumen epithelium, leading to LPS penetration into the circulatory system and distribution to various tissue organs, thereby triggering a systemic inflammatory response [6].

Rumenitis is one of the outcomes resulting from SARA; this SARA-associated rumen epithelium damage further increases ruminal LPS entering the blood circulation, resulting in a worsened situation [7,8]. The key to effectively combating SARA is to understand the inflammatory response mechanism on the rumen epithelium and to develop additives or drugs with anti-inflammatory effects. Rumen inflammation in SARA cows is initiated by the lysis and proliferation of gram-negative bacteria, which releases endotoxins into the rumen environment [9]. As a well-known ligand of toll-like receptor 4 (TLR4), LPS can interact with the TLR4 and activate diverse inflammatory pathways via nuclear factor kappa B (NF-κB) and mitogen-activated protein kinase (MAPK) mediation. [10]. As a result, it leads to the expression of proinflammatory mediators, such as tumor necrosis factor-α (TNF-α), interleukin-1β (IL-1β) and interleukin-6 (IL-6) [11]. Inflammation that occurs topically in the rumen harms the absorption and barrier functions of the rumen epithelium, which allows cows to have increased absorption of harmful compounds, causing inflammation in other organs, such as liver abscesses, mastitis, laminitis, and then progressing to a more severe systemic inflammatory response [8,12,13,14]. Therefore, effective anti-inflammatory agents are urgently needed to alleviate or block this process to alleviate SARA.

Propionic acid, classified as one of the short-chain fatty acids (SCFAs), has been documented to possess anti-inflammatory properties [15,16]. It is produced through the fermentation of cellulose in feed by rumen microbiota during ruminant digestion [17]. Propionic acid can also be further transformed into glucose in the liver by gluconeogenesis as one of the energy sources for ruminants [18,19]. Propionic acid is a widely used feed additive due to its excellent antifungal properties and safety [20]. Moreover, a certain concentration of propionic acid can regulate rumen development in calves [21]. Not only that, it has been discovered that propionic acid possesses anti-inflammatory properties in recent studies [22,23]. It is reported that sodium propionate (SP) could reduce TNF-α, IL-6 and IL-1β levels in a dose-dependent manner within the mammary gland of mice [24]. Several studies have reported that SCFA inhibits NLRP3 inflammasome activation induced by NF-κB and LPS, and also inhibits autophagy. [25,26]. However, the anti-inflammatory mechanism of propionic acid is well-studied in mice, but not in cows. Therefore, the hypothesis that propionate can alleviate the LPS-induced inflammation of the in rumen epithelial cells (RECs) by inhibiting the NF-κB and MAPK pathways was proposed. The objective of this research was to explore the role of SP in the inflammatory response of rumen epithelium through primary RECs culture assay confirmed whether SP exerted its effect on NF-κB and MAPK, and established an inflammatory model of RECs, which provided a method and basis for the study of the mechanism of SARA-induced rumenitis.

## 2. Results

### 2.1. Culture and Identification of Rumen Epithelial Cells

The RECs were isolated and cultured as indicated in the prescribed methodology, with the corresponding outcomes depicted in Figure 1A. The cells began to adherent after 24 h of incubation and appeared as a round sphere. After 48 h of incubation, the volume increased, and almost all cells were completely adherent. At 72 h, the cell morphology began to transform into the adherent state of epithelial cells and proliferate rapidly after culturing for 96 h.

ACTB (Beta Actin) and cytokeratin18 (CK18) protein expression profiles were identified with immunofluorescence. The results of cell immunofluorescence are shown in Figure 1B. DAPI was used to label the nuclei, which were colored blue; ATCB was stained green and CK18 was stained red. The results illustrated that the isolated and cultured cells stably expressed the above proteins, indicating that the RECs were successfully isolated and cultured, which can meet the requirements of subsequent tests.

### 2.2. Effects of LPS, SP and PDTC on Cell Viability

To explore the effects of LPS (1, 5, 10 μg/mL), SP (15, 25, 35 mM) and PDTC (10 μM) in RECs proliferation, the CCK-8 method was employed to assess the cellular viability following treatment with varying concentrations. As shown in Figure 1C, none of these three substances obviously affected cell viability (*p* > 0.05) and could be used in subsequent trials.

### 2.3. Establishment of LPS-Induced Inflammatory Rumen Epithelial Cells Models In Vitro

To explore the optimal action time of LPS, RECs were treated at 0, 1, 3, 6, 9 and 18 h, a total of six different time points, and observed for changes in the expression of TLR4, MyD88 and other key proteins of NF-κB and MAPK pathway (Figure 2A). As can be seen through Figure 2B, compared with the 0 h, the expression of TLR4 was extremely increased in a time-dependent manner at 1, 3, 6 and 9 h (*p* < 0.01), reached the highest level at 9 h and subsequently decreased, but the expression was still significantly increased at 18 h compared with 0 h group (*p* < 0.05). The expression of MyD88 showed the same trend as TLR4, peaking at 9 h, and the expression at 18 h decreased compared with 9 h (Figure 2C). NF-κB P65 protein began to be significantly phosphorylated at 6 h (*p* < 0.01) uniformly; the phosphorylation of IκB reached the highest level at 6 h (*p* < 0.01) (Figure 2D,E). JNK in the MAPK signaling pathway was significantly (*p* < 0.01) phosphorylated beginning at 3 h and peaking at 9 h, consistent with a trend toward TLR4. After, ERK was more phosphorylated at 1–18 h compared with 0 h (*p* < 0.01). The content of P-p38 reached a peak at 6 h (*p* < 0.01). After analyzing the expression changes of each key protein, we determined that the optimal stimulation time of LPS was 6 h.

To explore the optimal concentration of LPS, RECs were stimulated with 0, 1, 5 and 10 μg/mL LPS (Figure 3A). The protein gray analysis results showed that, at the concentrations of 1, 5 and 10 μg/mL of LPS, the contents of TLR4 and MyD88 both exhibited extremely significant elevation in comparison to the 0 μg/mL group (*p* < 0.01), and the highest value occurred when the concentration of LPS was 5 and 10 μg/mL (Figure 3B,C). Moreover, in pathway NF-κB, the phosphorylation of p65 protein showed a highly significant increase at LPS concentrations of 10 μg/mL (*p* < 0.01). Similarly, P-IκB also reached its maximum at 10 μg/mL (*p* < 0.01) (Figure 3D–E). According to the above results, 10 μg/mL of LPS was the best concentration and was used to stimulate the RECs for immunofluorescence.

To determine the effect of LPS stimulation on RECs to establish an inflammatory model by immunofluorescence assay. As shown in Figure 3F, P-NF-κB p65 is red-stained in cells, and the nucleus is blue-stained. Upon stimulation of RECs with LPS at a concentration of 10 μg/mL for 6 h, there was a significant increase in the amount of P-NF-κB p65 translocating to the nucleus as opposed to the control group, which was consistent with the WB results.

In addition, we studied the effects of LPS at different concentrations on the MAPK pathway. As depicted in Figure 4, the levels of P-JNK were significantly increased compared with those of the control group when the concentrations of LPS were 1, 5 and 10 μg/mL (*p* < 0.01), the levels of P-ERK were significantly increased when the concentrations were 5 and 10 μg/mL (*p* < 0.01) and the levels of P-p38 MAPK were increased when the concentration of LPS was 10 μg/mL (*p* < 0.01).

### 2.4. Changes of Cytokines in Rumen Epithelial Cell Inflammation Models

To determine the impact of varying concentrations of LPS on cytokine release and mRNA expression in RECs, it can be seen in Figure 5A,B,D,E that the contents and mRNA expressions for TNF-α and IL-1β demonstrated a marked increase in significance when exposed to LPS concentrations of 1, 5 and 10 μg/mL as compared to the control group (*p* < 0.01). IL-6 was highly increased at LPS concentrations of 5 and 10 μg/mL (*p* < 0.01), and mRNA expression was increased at 1, 5 and 10 μg/mL (*p* < 0.05 or *p* < 0.01) (Figure 5C,F). This is consistent with the previous results; LPS causes inflammatory signaling pathways to be activated, releasing cytokines. It was further confirmed that the RECs inflammation model was established successfully.

### 2.5. Effects of Different Concentrations of SP on LPS-Induced NF-κB and MAPK Signaling Pathway Proteins

WB results presented in Figure 6A–C when the concentrations of SP were 25 and 35 mM, the amount of P-NF-κB p65 and P-IκB protein expression were extremely significantly reduced (*p* < 0.01), indicating that SP effectively inhibited the activation of NF-κB inflammatory signaling pathway. The results of the immunofluorescence assay depicted in Figure 6 further support this conclusion, with SP (35 mM) reversing the increased nuclear incorporation of P-NF-κB p65 observed after LPS (10 μg/mL) stimulation. The results depicted in Figure 7 illustrate a significant increase in the levels of P-JNK, P-ERK and P-p38 MAPK, after LPS stimulation (10 μg/mL) (*p* < 0.01). However, there was a significant decrease in the aforementioned levels, upon administering SP with concentrations of 15, 25 and 35 mM (*p* < 0.01). To summarize, the present study reveals that SP can impede the activation of NF-κB and MAPK inflammatory signaling pathways triggered by LPS within RECs.

### 2.6. Effects of SP on LPS-induced Cytokine Release and mRNA Expression

Upon treating the cells with a concentration of 10 μg/mL of LPS, there was a significant increase in the levels of TNF-α, IL-1β and IL-6 as compared to the control group (*p* < 0.01). Upon supplementing the treatment with 35 mM SP, significantly decreased levels of TNF-α and IL-1β were observed (*p* < 0.05) (Figure 8A–C). The mRNA expressions of TNF-α, IL-1β and IL-6 in response to treatment with LPS alone were extremely increased compared with the control group (*p* < 0.01), and those of TNF-α with the addition of SP treatment were significantly decreased (*p* < 0.01), as well as those of IL-6 and IL-1β were decreased (*p* < 0.05) (Figure 8D–F). The above results support that SP effectively alleviated the inflammation of RECs induced by LPS.

## 3. Discussion

Nowadays, SARA has become a common metabolic disease, and its incidence is between 19% and 26% according to multiple surveys [27,28]. Inappropriate feeding practices lead to the buildup of volatile fatty acids within the rumen, creating an acidic environment that disrupts the buffering system, which is the leading cause of SARA [29,30,31]. The accepted definition of SARA is a period of duration between acute and chronic, during which ruminal pH is less than 5.5 and higher than 5.2 [32,33]. The current clinical diagnostic modality of SARA mainly relies on single-point ruminal pH measurements; however, the sensitivity and accuracy of this modality are not satisfactory [34]. More regrettably, many clinical symptoms appear delayed, which makes SARA more difficult to diagnose, so the actual prevalence of this disease may be even higher [34,35,36]. SARA can cause complications, such as rumen mucosal damage, rumenitis, laminitis and liver abscesses, which lead to decreased production performance and significant economic losses for the dairy industry. [8,37]. Research investigations in the United States have shown up to USD 500 million to USD 1 billion in losses caused by the disease [35]. Cows with SARA often have elevated concentrations of free LPS in the rumen, which triggers rumenitis as a proinflammatory mediator [8,38]. Not only that, once the rumen epithelium is inflamed, its barrier function would be impaired, which will increase the chance of pathogenic substances, including LPS, entering the blood circulation, which affects other organs [12,39,40]. Propionic acid has well-established anti-inflammatory effects as one of the SCFA family members, but its mechanism of action in rumen epithelial cells remains unknown [41]. In this study, an in vitro inflammatory cell model was established by LPS stimulation of RECs, through which the anti-inflammatory mechanism of SP in RECs was explored, providing a potential therapeutic strategy and research basis for alleviating SARA-induced inflammation.

LPS is a significant element of the outer membrane of Gram-negative bacteria, which can elicit inflammation and cytokine production in animal models, such as mice and cattle. [42,43]. Because of this, LPS is widely used in establishing cellular inflammatory models, which is also the most important inducer of rumenitis during SARA and multi-organ inflammation in dairy cows [8]. In this experiment, LPS with a concentration of 10 μg/mL was used to establish the model; however, there were apparent differences in this concentration among the cells of different species genera. As an illustration, it has been observed that guinea pig gastric mucosal cells exhibit a remarkable sensitivity to lipopolysaccharides (LPS). In particular, exposure to *H. pylori* LPS (>1 ng/mL) and *E. coli* LPS (>1 ng/mL) for a duration of 8 h results in the induction of LPS in these cells [44]. Such heightened sensitivity in guinea pigs may be attributed to their predisposition towards sensitization. It has been documented that porcine intestinal epithelial cells, which are physiologically close to human cells, can reduce their cell viability when stimulated with 0.25 µg/mL LPS [45], whereas murine mammary epithelial cells require 1 µg/mL LPS [42], all at lower stimulus concentrations than those used in this experiment. Notably, the use of LPS to model inflammation in goat rumen epithelial cells has also been reported at a concentration of 1 µg/mL for 24 h, which may be related to the different origins and environments of the cells [46]. LPS content within the ruminal fluid of animals with SARA is also often increased [47]. In an induced SARA experiment, cows’ ruminal LPS increased from 9.1 and 9.3 μg/mL to 143.3 and 172 μg/mL, which were much higher than the concentrations used in our experiment [48]. However, in another six-week SARA challenge, the concentration of free LPS in rumen fluid ranged from 28,184 to 107,152 endotoxin units (EU)/mL (equivalent to 2.8 to 10.7 µg/mL), while in the other eight-week test, the concentration of free LPS in rumen fluid ranged from 30,768 to 130,589 EU/mL [7,8] (equivalent to 3.1 to 13.0 µg/mL), These above reports provide a reference for the LPS stimulation concentration used in our experiment. Because RECs were directly exposed to the stimulation of LPS in this experiment, a lower concentration of 10 μg/mL is considered. Besides, it was verified by the LPS concentration gradient (1, 5, 10 μg/mL) stimulation test, which concluded that 10 μg/mL was not only the concentration that was the closest to SARA but also the optimal concentration to induce inflammation of RECs in vitro.

In the context of SARA in dairy cows, the degradation of the rumen environment results in an elevated level of LPS within the rumen epithelium, eliciting an inflammatory response [2]. MAPK and NF-κB are the two major inflammatory signaling pathways in the body, and TLR4 is the upstream regulator of these two major pathways [49,50]. Upon binding to TLR4, LPS released by bacteria activates TLR4 and MyD88 [47]. In this experiment, the expression of TLR4 and MyD88 increased significantly in 6–9 h after LPS stimulation, which indicates the upstream pathway of NF-κB is activated. The binding of TLRs to MyD88 activates the IκB kinase complex, which phosphorylates and degrades IκB, leading to NF-κB translocation into the nucleus where it induces other inflammatory cytokines [51]. The results of the present study demonstrated that the phosphorylation levels of p65 and IκB were significantly increased (*p* < 0.01) and that p65 nuclear import was significantly increased after RECs were stimulated with LPS. Meanwhile, the contents of TNF-α, IL-6 and IL-1β in cell supernatant were significantly increased (*p* < 0.01), which also further verified the result that NF-κB inflammatory signaling was activated. Zhang et al. showed that the mRNA levels of IL-1β, IL-2, IL-6 and IL-8 were significantly increased after LPS treatment of RECs, which was similar to our results, but the stimulation time was consumed for a longer period of 24 h [52]. The MAPK signaling pathway has been linked to the initiation of inflammatory factors, such as the activation of p38, ERK and JNK [53]. According to the results obtained, phosphorylation of JNK exhibited its most prominent activation between 6 and 9 h, aligning with the activation pattern of TLR4 and MyD88. On the other hand, p38 demonstrated its strongest phosphorylation at the 6-h mark, when the LPS concentration was 10 μg/mL. Furthermore, the phosphorylation of ERK and p38 reached their maximum levels during this phase, indicating the activation of the MAPK signaling pathway. As the activity of MAPKs increases, especially p38 MAPK, inflammatory mediators are also enhanced at the transcriptional and translational levels to enhance the inflammatory response [54]. It is well documented that stimulation of RECs for 6 h using LPS (10 µg/mL) can activate the MAPK pathway [43], consistent with our results, while GRECs (goat rumen epithelial cells) required LPS (5 µg/mL) stimulation for 6 h at a lower dose to activate [55]. The results of the experiment indicate the activation of the NF-κB and MAPK signaling pathways through p65 translocation into the nucleus following LPS stimulation. This led to an increase in cytokines in RECs, thereby validating the successful establishment of the RECs inflammation model.

Carbohydrate fermentation within the rumen of ruminants is a vital biochemical process that results in the production of key compounds, such as acetate, propionate and butyrate, along with hydrogen and carbon dioxide [56]. Butyrate, which belongs to the SCFA family, has been extensively researched for its anti-inflammatory and anti-carcinogenic properties, whereas our comprehension of propionate remains insufficient. [15,57]. Propionate in the rumen serves as the primary substrate for ruminant gluconeogenesis. In addition to its role in gluconeogenesis, propionate will also participate in other physiological processes, such as promoting intestinal epithelial cells’ vitality and the autophagy of liver cells [58,59]. In the dairy industry, propionate is gradually used as a feed additive, which could mitigate the negative energy balance (NEB), prevent milk fever and enhance rumen development [20]. Several studies have indicated that propionic acid possesses anti-inflammatory properties in dextran sulfate sodium (DSS)-induced colitis mouse model and carrageenan (CAR)-induced rat paw inflammation [41,60]. Wang et al. proposed that sodium propionate can prevent LPS-induced mouse mastitis by inhibiting NF-κB inflammatory signal pathway [24]. In addition, Kara et al. showed that infusion of calcium propionate in the first week after delivery of cows can significantly reduce the incidence rate of metritis, but its specific mechanism has not been clearly described [61]. Drawing upon the aforementioned research, our hypothesis posits that the administration of SP may be an effective means of ameliorating the inflammation of the rumen epithelium in cows afflicted with SARA by impeding the NF-κB and MAPK signaling pathways. In this study, RECs were treated with 15, 25 and 35 mM SP after LPS stimulation to assess the effect of different concentrations of SP on the NF-κB and MAPK pathway. The levels of P-NF-κB p65 and P-IκB were significantly decreased when the concentration of propionate was 25 and 35 mM (*p* < 0.01). Wang et al. used 1 mM SP in the experiment to also significantly inhibit the phosphorylation level of p65 and IκB. The SP concentration was significantly lower than that in this experiment, possibly because mouse mammary epithelial cells were more sensitive to SP [24]. Huang et al. also proved that SP can inhibit the P-NF-κB p65 in type 2 diabetes-induced inflammation, but it has little effect on IκB [62]. Moylan et al. also confirmed our results, which showed that SP could inhibit the activation of MAPK and NF-κB signaling pathway in vitro model of preterm birth [63]. However, Park et al. proposed that SP increased the phosphorylation level of p38 MAPK, which induced cell apoptosis. The reason may be that a high concentration of SP (20 mM) was used to treat breast cancer cells in mice, which was much higher than other mouse cell experiments [64]. In this experiment, RECs were treated with 35 mM SP, and immunofluorescence showed that the intensity of red-stained P-NF-κB p65 protein was significantly reduced, indicating that SP can inhibit p65 translocation to the nucleus so that downstream target gene transcription. Dissociation of NF-κB p65 dimers into the nucleus promotes the release of cytokines, which is a major indicator reflecting inflammation [65]. Consequently, cytokine secretion was maximally inhibited when the SP concentration was 35 mM. The results that SP can inhibit the production of proinflammatory cytokine have also been verified in previous experiments [64,66,67].

The conducted experiment serves to validate that SP demonstrates anti-inflammatory properties within RECs in a model of LPS-induced inflammation by modulating NF-κB and MAPK pathways. Many studies have shown that the range of SP concentration in the rumen is 15–25 mM, while 35 mM is already the SP concentration of dairy cows afflicted with SARA [7,8]. Although the increase of SP content will reduce the pH of the rumen fluid, the acidity of lactic acid, one of the components of rumen fluid, is 10 times higher than that of SP, which has a greater effect on the pH reduction of rumen fluid during SARA [68]. The application of SP would inhibit the rumenitis induced by LPS and alleviate or prevent the development of SARA. Nevertheless, a limited number of research works have been conducted to examine the anti-inflammatory impact of SP in the rumen, and its underlying mechanism remains enigmatic. This research was only conducted through RECs in vitro but not verified in vivo, which is also a deficiency of this study. Consequently, it is crucial to conduct additional investigations into the anti-inflammatory mechanism of SP in the rumen epithelium, as well as to determine the ideal application concentration in vivo. Such studies will offer valuable treatment options and a sound theoretical foundation for addressing rumenitis in cows afflicted with SARA.

## 4. Conclusions

In summary, the present study indicates that LPS can activate inflammatory signaling pathways in the RECs, while SP effectively shields RECs against inflammation caused by LPS stimulation. This is attributed to SP suppressing LPS-driven activation of the NF-κB/MAPK pathway, thus leading to the downregulation of cytokine production. These results reveal the mechanism of signal transduction associated with the inflammatory response in the rumen epithelium cells, which also provide potential therapeutic targets and drugs for local inflammation in the rumen epithelium provoked by SARA and accompanying systemic inflammatory response.

## 5. Materials and Methods

Animal Ethical and Welfare Committee, Northwest A&F University approved the experimental designs and protocols (Approval No.2021048), which were executed in accordance with the university’s guidelines (NWAFAC1008).

### 5.1. Animals and Tissue Collection

Cows were obtained from a commercial dairy farm with more than 10,000 cows in Baoji City, Shaanxi Province, China (34.33548N, 108.71542E). The abdominal cavity of newborn Holstein calves weighing 35–45 kg (n = 5) was opened under anesthesia in a sterile condition by surgical veterinarians. The rumen tissues were collected at the ventral sac, and washed three times with 4 °C precooling sterile saline and soaked, then sent to the laboratory within an hour.

### 5.2. Isolation, Culture and Identification of Primary Rumen Epithelial Cells (RECs)

Rumen epithelium was separated by ophthalmic micro-tweezers on the sterile operation table. The technique of isolating and culturing RECs has already been described in detail [69]. In brief, tissues were washed three times with 4 °C sterile saline and shredded with ophthalmic scissors. Small pieces of tissue were washed twice, first with PBS solution containing penicillin (2500 U/mL) and streptomycin (2500 U/mL), and subsequently with PBS solution containing amphotericin B (1000 U/mL) and gentamicin (6 µg/mL) in a gas bath thermostatic oscillator (THZ-82A, SAIDELISI, Tianjin, China). The minced laminae were digested with 0.25% trypsin (T8150, Solarbio Science & Technology, Beijing, China); then, the digestion was terminated with fetal bovine serum (FBS, FB15015, Clark Bioscience, Richmond, VA, USA). The digestion fluid with RECs was centrifuged (1000×rpm, 10 min at 4 °C) to remove the residual trypsin. A solution containing 15% FBS and low-glucose DMEM SH30021.01B, HyClone, Logan, OH, USA) was used to resuspension the cells after centrifugation. After that, the cells were sieved through a 40 µm cell strainer (CLS431750, Corning, New York, NY, USA). The viability of cells was determined using trypan blue (C0040, Solarbio Science & Technology, Beijing, China) exclusion. To count the cells, a hemocytometer was used to adjust cell density. Cells were cultured in six-well plates at a density of 1 × 10^6^ cells/mL and incubated at 37 °C in a humidified incubator containing 5% CO_2_. (150i, Thermo Fisher Scientific, Waltham, MA, USA). Primary RECs were cultivated in DMEM medium, which was replaced every other day. Primary RECs obtained by the above steps were harvested on the sixth day. The primary rumen epithelial cells were utilized in all of the following cell experiments, and no passage operations were conducted. RECs were seeded in 24-well culture plates, which were plated with cell-climbing slices for identification. Cells were fixed with 4.0% formaldehyde for 30 min, permeabilized with 0.1% Triton X-100 and then incubated overnight with the primary antibody ACTB. (1:400; Cat. 20536-1-AP, Proteintech, Wuhan, China) and CK18 (BM0032, Boster Biological, Wuhan, China) after washing by PBS. Cells were incubated with secondary antibody: FITC labeled goat anti-rabbit IgG (A0562, Beyotime, Shanghai, China) and Alexa Fluor^®^ 647 labeled goat anti-mouse IgG (1:400; Cat. ab150115, Abcam, Cambridge, UK); then, nucleus were stained with DAPI (C1006, Beyotime, Shanghai, China). Thereafter, the cells were added antifade mounting medium (P0126, Beyotime, Shanghai, China) and observed under a fluorescence microscope (Carl Zeiss, Jena, Germany).

### 5.3. Cell Viability Assay

To perform the cell toxicity assay, using counting kit-8 (CCK8; CA1210, Solarbio, Beijing, China), according to the manufacturer‘s direction, the cell was adjusted to 1 × 10^5^ cells/mL and seeded in 96-well plates. Cell medium was added 15, 25, 35 mM SP (P5436, Sigma-Aldrich, St Louis, MS, USA) and 10 μM pyrrolidine dithiocarbamate (PDTC; P8765, Sigma-Aldrich, St Louis, MS, USA), then incubated for 2 h. The cells were stimulated by 1, 5 and 10 μg/mL LPS (L4391, from the *E. coli* 0111:B4, Sigma-Aldrich, St Louis, MS, USA) for 6 h. After this, 10 μL CCK8 solution was added to each well and then the cells were incubated for another 3 h. The OD value was measured at 450 nm by a microplate reader (Multiscan FC, Thermo Fisher Scientific, Waltham, MA, USA).

### 5.4. Cell Treatments

To building-up the inflammation model of RECs, we investigate the optimal time and concentration of LPS-induced inflammation. RECs were stimulated at different times (0, 1, 3, 6, 9, 18 h) and LPS (0, 1, 5, 10 μg/mL). After determining the optimal time and LPS concentration, RECs were treated with SP (15, 25, 35 mM) or 10 μM PDTC for 2 h in advance and subsequently processed together with LPS.

### 5.5. Protein Extraction and Western Blotting (WB) Assay

The protein extraction kit (P0013B, Beyotime, Shanghai, China) was applied to extract the proteins of RECs and then a BCA protein determination kit (P0012, Beyotime, Shanghai, China) was used to measure protein concentration. Proteins’ fractions were then separated on SDS-PAGE electrophoresis gels and transferred to PVDF membranes (IPVH00010, Millipore, Boston, MA, USA), which were blocked by using 5% skim milk powder for 2 h. Membranes were blocked, then incubated at 4 °C overnight with multiple primary antibodies, including TLR4, MyD88, P-NF-κB p65 (ab22048, ab2068, ab86299, Abcam, Cambridge, UK), NF-κB p65 (AF5006, Affinity Biosciences, Jiangsu, China), P-IκBα, IκBα, P-JNK, JNK, P-ERK, ERK, P-p38 MAPK, p38 MAPK (9246, 4814, 9251, 9252, 4370, 4695, 4511, 8690, Cell Signaling Technology, Danvers, MA, USA), and β-actin (c-47778, Santa Cruz, Dallas, TX, USA), respectively. After that, secondary antibodies were incubated on the membranes for 45 min, which were detected by enhanced chemiluminescence (ECL) system (5600, Tanon, Shanghai, China) and analyzed using the ImageJ software (Media Cybernetics, Bethesda, MD, USA).

### 5.6. Immunofluorescence

Cells were grown until reaching 90% confluency and then fixed using 4% paraformaldehyde solution for 20 min. Antigen was repaired by using EDTA-Na2 (1 mM, E8030, Solarbio, Beijing, China) at 95 °C for 5 min. Then the primary antibody against P-NF-κB p65 (1:200, ab86299, Abcam, Cambridge, UK) was applied to incubate the slides at 4 °C overnight. After that, the slides were incubated with goat anti-rabbit IgG conjugated with Cy3 (P0193, Beyotime, Shanghai, China) for 30 min and counterstained with Hoechst 33,258 (C1018, Beyotime, Shanghai, China). At last, the slides closed with an antifade mounting medium were examined under laser confocal microscopy.

### 5.7. Enzyme-Linked Immunosorbent Assay (ELISA)

To detect the concentration of inflammatory cytokines, the cell-free supernatant of RECs was harvested for ELISA assay by using a commercial ELISA kit (TNF-α: mL777163, 6.25–200 pg/mL; IL-1β: mL777027, 20–640 pg/mL; IL-6: mL064296, 6.25–200 pg/mL; Shanghai Enzyme-linked Biotechnology, Shanghai, China).

### 5.8. Quantitative Real-Time PCR

Total RECs mRNA was extracted using RNAiso Plus reagent (Code No. 9109, TaKaRa, Shiga, Japan), and mRNA concentrations were measured utilizing a NanoDrop (2000C, Thermo Fisher Scientific, Waltham, MA, USA) before reverse-transcribing into cDNA (SuperScript™ RT reagent kit, RR047A, TaKaRa, Shiga, Japan). The gene primers of inflammatory cytokines were applied from Zhao et al. (Table 1) [70]. Quantitative reverse transcription-polymerase chain reaction (qRT-PCR) was used to measure mRNA expression levels by using SYBR Green (MF787-01, Mei5bio, Beijing, China). Results were analyzed using the 2^−ΔΔ*C*T^ method.

### 5.9. Statistical Analysis

Statistical analysis was conducted using SPSS 19.0 software (Statistical Package for the Social Sciences, IBM, Chicago, IL, USA). One-way analysis of variance (ANOVA) was used to compare the data more than two groups. Bonferroni correction was used to perform multiple testing corrections. Results are expressed as means ± standard errors of the means (SEMs). *p* < 0.05 was statistically significant, and *p* < 0.01 was marked significant.

## Figures and Tables

**Figure 1 toxins-15-00438-f001:**
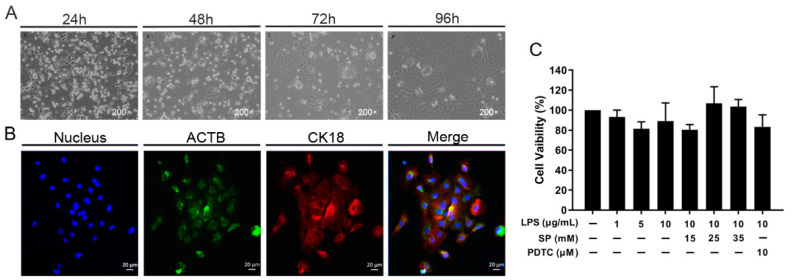
Culture, identification and viability test of RECs. (**A**) RECs in the culture at different times; (**B**) ACTB and CK18 protein expression in RECs; (**C**) the cellular viability following exposure to distinct concentrations of LPS (1, 5, 10 μg/mL), SP (15, 25, 35 mM) and PDTC (10 μM).

**Figure 2 toxins-15-00438-f002:**
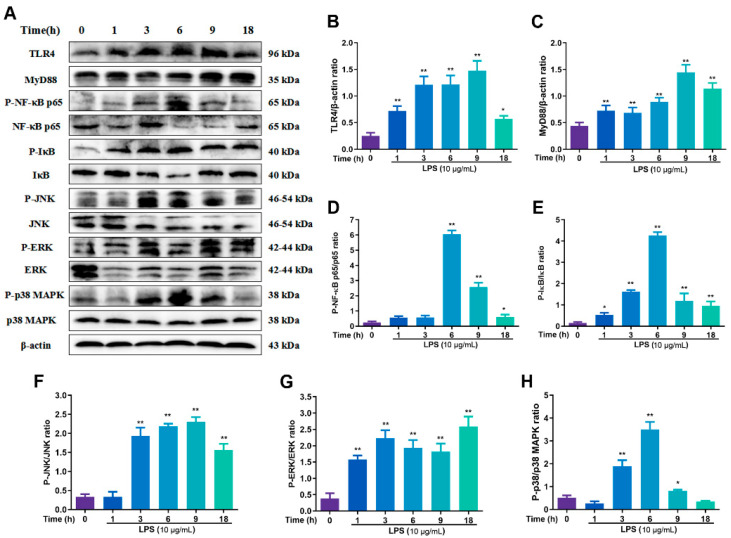
The changes in the expression of key proteins of NF-κB and MAPK signaling pathway after LPS stimulation in RECs at different times. (**A**) The WB results of TLR4, Myd88, P-NF-κB p65, NF-κB p65, P-IκBα, IκBα, P-JNK, JNK, P-ERK, ERK, P-p38 MAPK, p38 MAPK and β-actin; (**B**–**H**) The gray analysis of TLR4, Myd88, P-NF-κB p65, P-IκBα, JNK, ERK and p38 MAPK. * *p* < 0.05, ** *p* < 0.01, * was compared to 0 h group. N = 5 in each group.

**Figure 3 toxins-15-00438-f003:**
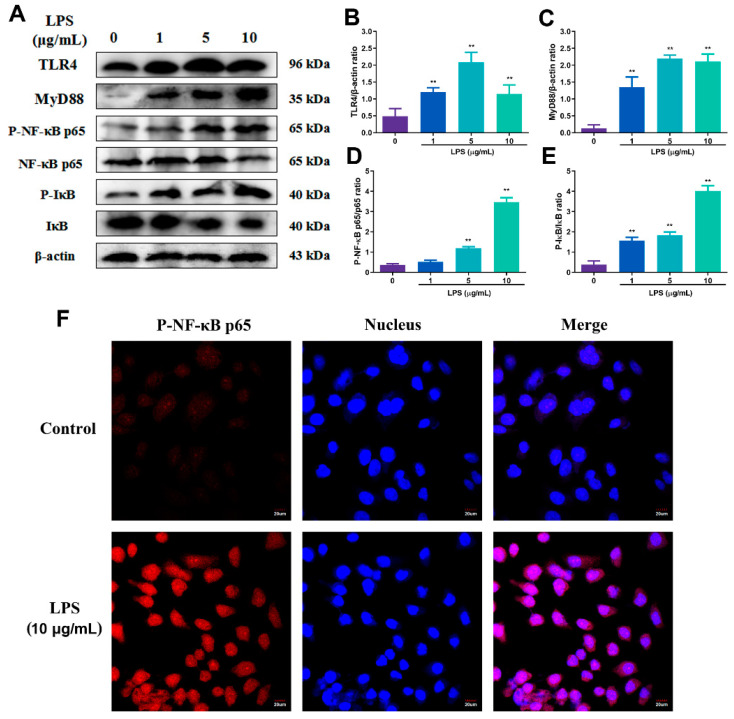
The changes in the expression of key proteins of NF-κB signaling pathway after LPS stimulation at different concentrations and P-NF-κB p65 cellular localization. (**A**) Effects of 0, 1, 5 and 10 μg/mL LPS on TLR4, Myd88, P-NF-κB p65, P-IκBα, NF-κB p65, IκBα protein expression of RECs; (**B**–**E**) the gray analysis of TLR4, Myd88, P-NF-κB p65, P-IκBα; (**F**) effect of 6 h and 10 μg/mL LPS on P-NF-κB p65 cellular localization in RECs. The red light represents P-NF-κB p65, and the blue light represents the nucleus. ** *p* < 0.01, * was compared to 0 μg/mL group. N = 5 in each group.

**Figure 4 toxins-15-00438-f004:**
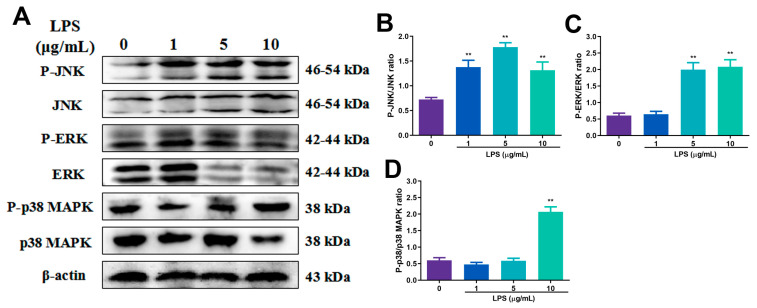
The changes in the expression of key proteins of MAPK signaling pathway after LPS stimulation at different concentrations. (**A**) Effects of 0, 1, 5, 10 μg/mL LPS on P-JNK, P-ERK, P-p38 MAPK, JNK, ERK, p38 MAPK and β-actin protein expression in RECs; (**B**–**D**) phosphorylation levels of JNK, ERK and p38 MAPK. ** *p* < 0.01, ** was compared to 0 μg/mL group. N = 5 in each group.

**Figure 5 toxins-15-00438-f005:**
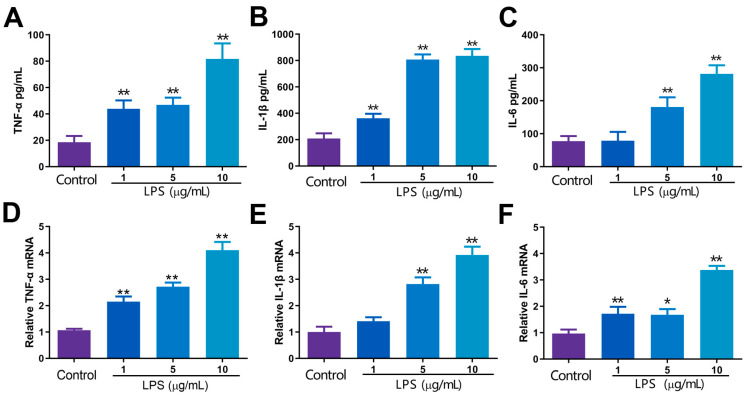
Effects of varying concentrations of LPS on the release of cytokines and mRNA expression. (**A**–**C**) Effects of 0, 1, 5, 10 μg/mL LPS on TNF-α, IL-1β and IL-6 content in RECs supernatant; (**D**–**F**) mRNA expression of TNF-α, IL-1β and IL-6 in RECs. * *p* < 0.05, ** *p* < 0.01, * was compared with control group. N = 5 in each group.

**Figure 6 toxins-15-00438-f006:**
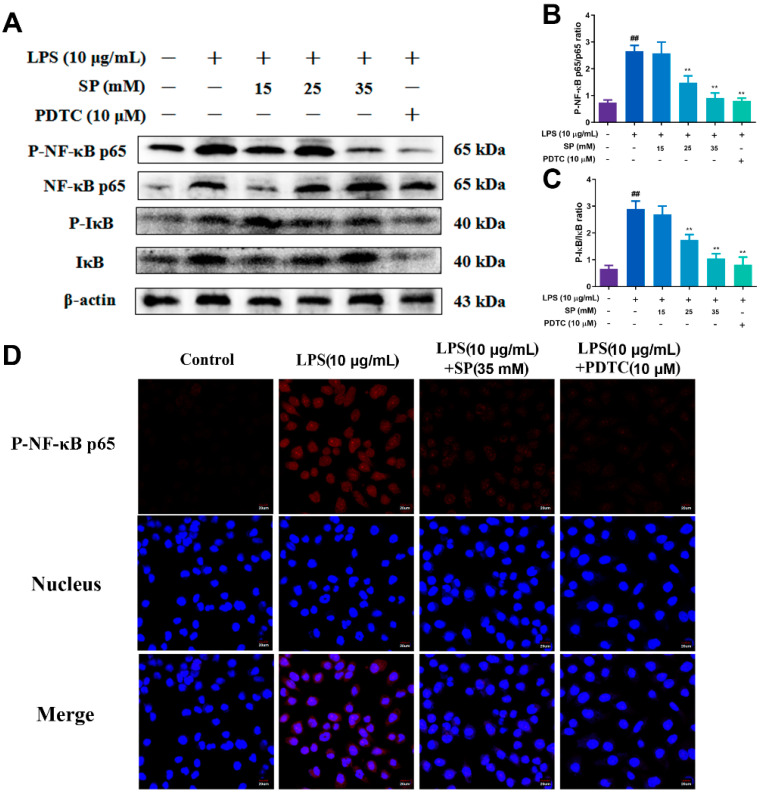
Effects of concentrations varying of SP on LPS-induced NF-κB signaling pathway in RECs and P-NF-κB p65 cellular localization. (**A**) WB protein bands of P-NF-κB p65, P-IκBα, NF-κB p65, IκBα in RECs treated with 10 μg/mL LPS and 15, 25, 35 mM SP and 10 μM PDTC; (**B**,**C**) phosphorylation intensities of NF-κB p65 and IκBα; (**D**) effect of 10 μg/mL LPS, 35 mM SP and 10 μM PDTC on P-NF-κB p65 cellular localization in RECs. The red light represents P-NF-κB p65, and the blue light represents the nucleus. ^##^
*p* < 0.01, ** *p* < 0.01, ^#^ was LPS group vs. control group, * was SP (15, 25, 35 mM) group or PDTC group vs. LPS group. N = 5 in each group.

**Figure 7 toxins-15-00438-f007:**
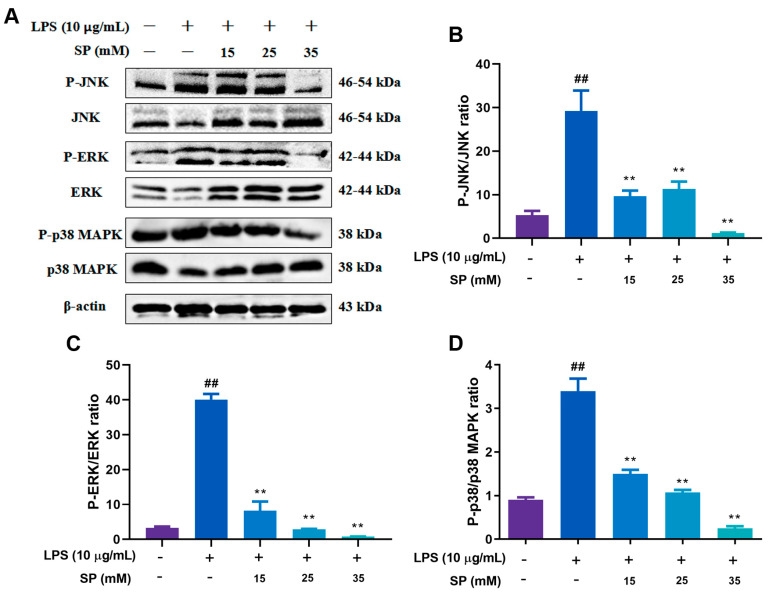
Effects of different concentrations of SP on LPS-induced MAPK signaling pathway proteins. (**A**) WB protein bands of P-JNK, P-ERK, P-p38 MAPK, ERK, JNK, p38MAPK and β-actin in RECs treated with 10 μg/mL LPS and 15, 25, 35 mM SP; (**B**–**D**) the phosphorylation intensities of JNK, ERK and p38MAPK. ^##^
*p* < 0.01, ** *p* < 0.01, ^#^ was LPS group vs. control group, * was SP (15, 25, 35 mM) group vs. LPS group. N = 5 in each group.

**Figure 8 toxins-15-00438-f008:**
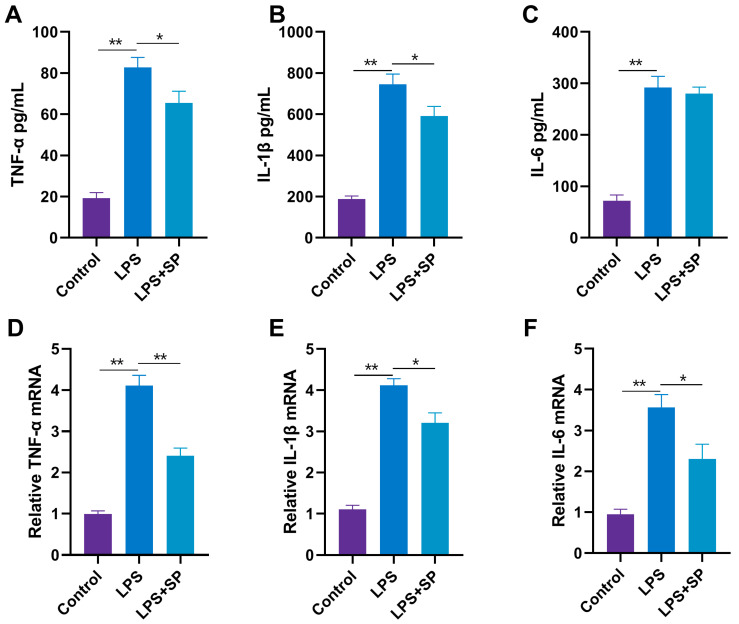
Effects of SP on LPS-induced cytokine release and mRNA expression. (**A**–**C**) Effects of 10 μg/mL LPS and 35 mM SP on TNF-α, IL-1β and IL-6 content in RECs supernatant; (**D**–**F**) mRNA expression of TNF-α, IL-1β, and IL-6 in RECs. * *p* < 0.05, ** *p* < 0.01. N = 5 in each group.

**Table 1 toxins-15-00438-t001:** The primers sequences of the genes.

Gene	Primer Sequences (5′-3′)	Tm (°C)	Length
TNF-α	For CTGCCGGACTACCTGGACTAT Rev CCTCACTTCCCTACATCCCTAA	60.75 58.35	234 bp
IL-6	For AACGAGTGGGTAAAGAACGC Rev CTGACCAGAGGAGGGAATGC	58.49 59.82	140 bp
IL-1β	For CTGAACCCATCAACGAAA Rev ATGACCGACACCACCTGC	52.48 59.65	190 bp
β-actin	For GCCCTGAGGCTCTCTTCCA Rev GCGGATGTCGACGTCACA	60.99 60.13	101 bp
GAPDH	For CCTGCCAAGTATGATGAGAT Rev AGTGTCGCTGTTGAAGTC	58.53 59.75	117 bp

For: Forward; Rev: Reverse.

## Data Availability

Data sharing not applicable.

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
