# Peer review of "Sodium Propionate Relieves LPS-Induced Inflammation by Suppressing the NF-ĸB and MAPK Signaling Pathways in Rumen Epithelial Cells of Holstein Cows"

_toxins, 2023, doi:10.3390/toxins15070438_

Round 1
Reviewer 1 Report
This study explores sodium propionate's role in mitigating LPS induced inflammatory response of rumen epithelium. Overall, the article is well constructed, the experiments were well conducted, and the analysis was correctly performed. I have a few questions/concerns that need explanation.
1. Line 400; what is the size of the mesh? was it 300 microns? Correct it accordingly.
2. Primary RECs tend to grow slowly. What was the doubling time of the cells? How long did it take to reach 90% confluency?
3. Author should provide the passage number of RECs. Was the same passage number used for all the assays? if not, provide the relevant information for each assay.
4. Line 428; The bacterial source of LPS must be provided. Was it from E.coli or Pseudomonas?
5. Like the optimal time and conc. of LPS was determined prior to assays, why did authors choose to use SP just for 2 h? Was the time period for SP treatment optimized earlier? if yes, then please describe the results in a concise manner.
6. Why was just one reference gene (beta-actin) used for qPCR as housekeeping? Did it stably expressed across the treatments? The LPS challenge affects the stability of the generally well-known housekeeping genes. So 2-3 reference genes must be tried before selecting the best stable reference gene for qPCR analysis.
English quality is fine.
Reviewer 2 Report
Sodium propionate relieves LPS-induced inflammation by suppressing the NF-ĸB and MAPK signaling pathways in rumen epithelial cells of Holstein cows
toxins-2429937
Appreciable authors,
Through the review of the manuscript, I can appreciate the effort in the quality of the research. However, it is important to make some pressure and improvement within your manuscript, in this sense, allow me to request some changes. Authors must review in detail the instructions to the authors; the special references must be numbered in order of appearance in the text (including table captions and figure legends) and listed individually at the end of the manuscript. In the text, reference numbers should be placed in square brackets [ ], and placed before the punctuation; for example [1], [1–3], or [1,3]. https://www.mdpi.com/journal/toxins/instructions.
Additionally, please focus on all references included in the manuscript and please, it will be read according to the instruction to the authors (References should be described as follows, depending on the type of work:…. https://www.mdpi.com/journal/toxins/instructions.)
Additionally, some changes are necessary to the version of the manuscript, please find your highlighted text in the PDF file attached. Please focus-on on the addition or deletion spaces between characters highlighted along the manuscript
Figure 1B
It says: ATCB
It should say: ACTB
Figure 2B-H
It says: LPS (10mg/mL)
It should say: (10 mg/mL)
Figure 3F
It says: LPS (10mg/mL)
It should say: (10 mg/mL)
Line: 170
It says: group. (P < 0.01).
It should say: group (P < 0.01).
Figure3D
It says: LPS (10mg/mL)
It should say: LPS (10 mg/mL)
It says: +SP(35mM)
It should say: +SP (35 mM)
It says: +PDTC(10mM)
It should say: +PDTC (10 mM)
Line: 237
It says: the disease[35].
It should say: the disease [35].
Line: 244, 277, 339, 262, 265
It says: in vitro
It should say: in vitro
Line: 256
It says: to H. pylori LPS (> 1 ng/mL) and E. coli LPS (> 1 ng/mL)
It should say: to H. pylori LPS (> 1 ng/mL) and E. coli LPS (> 1 ng/mL)
Line 296
It says: JNK. [53].
It should say: JNK [53].
In the material and methods section, it is necessary that the authors include the catalog number, brand or name of the manufacturer, city of the manufacturer, and country of the manufacturer for all the equipment, reagents, and antibodies used. The authors will be able to identify within the pdf file attached (highlighted text), where the information is required.
I hope that these suggestions will allow you to contribute to increasing the quality of your manuscript.

Round 2
Reviewer 2 Report
Dear authors
Thaks you by attended all sugestions